



# Development and Application of a Supervised Pattern Recognition Algorithm for Identification of Fuel-Specific Emissions Profiles

Christos Stamatis[1] and Kelley C. Barsanti[1]

[1]Department of Chemical and Environmental Engineering and College of Engineering – Center for Environmental Research and Technology (CE-CERT), University of California, Riverside, Riverside, CA, USA

**Correspondence:** Kelley C. Barsanti (kbarsanti@engr.ucr.edu)

**Abstract.** Wildfires have increased in frequency, duration and size in the western United States (U.S.) over the past decades. These trends are projected to continue, with negative consequences for air quality across the U.S. Wildfires emit large quantities of particles and gases that serve as air pollutants and their precursors, and can lead to severe air quality conditions over large spatial and long temporal scales. Characterization of the chemical constituents in smoke as a function of combustion conditions,

fuel type, and fuel component is an important step towards improving the prediction of air quality effects from fires and evaluating mitigation strategies. Building on the comprehensive characterization of gaseous non-methane organic compounds (NMOCs) identified in laboratory and field studies, a supervised pattern recognition algorithm was developed that successfully identified unique chemical speciation profiles among similar fuel types common in western coniferous forests. The algorithm was developed using laboratory data from single fuel species and tested on simplified synthetic fuel mixtures. The fuel types

in the synthetic mixtures were differentiated but as the relative mixing proportions became more similar, the differentiation became poorer. Using the results from the pattern recognition algorithm, a classification model based on linear discriminant analysis was trained to differentiate smoke samples based on the contribution(s) of dominant fuel type(s). The classification model was applied to field data and despite the complexity of contributing fuels, and the presence of fuels "unknown" to the classifier, the dominant sources/fuel types were identified correctly. The pattern recognition and classification algorithms are

a promising approach for identifying the types of fuels contributing to smoke samples and facilitating selection of appropriate chemical speciation profiles for predictive air quality modeling, using a highly reduced suite of measured NMOCs. Utility and performance of the pattern recognition and classification algorithms can be improved by expanding the training and test sets to include data from a broader range of single and mixed fuel types.

## 1   Introduction

Research has showed that the western U.S. has seen an increase in the frequency and intensity of wildfires, over the last three decades (Jaffe et al. (2020), Miller et al. (2009) and Dennison et al. (2014)), which is projected to continue in the coming



decades (Westerling et al. (2006), Miller et al. (2009) and Dennison et al. (2014)). One of the consequences of wildfires is extremely poor air quality (McMeeking et al. (2005), McKenzie et al. (2006), Park et al. (2006), and Hu et al. (2018)).

Emissions from wildfires include carbon monoxide (CO), carbon dioxide (CO2), and methane (CH4); several hundreds of gas-phase non-methane organic compounds (NMOCs); and particulate matter (PM). While CO2 and CH4 are important greenhouse gases, NMOCs are of particular importance in the context of air quality because they serve as precursors to secondary air pollutants including photochemical ozone (O3) and secondary organic aerosol (SOA) (Alvarado and Prinn (2009)). The latter of which, SOA, is a major constituent of atmospheric PM (Zhou et al. (2017), Tomaz et al. (2018), Theodoritsi and Pandis

(2019)). In order to predict the air quality impacts of wildfires, differences in emissions and their effects on chemistry and pollutant formation must be represented in models (Kochanski et al. (2015), Pavlovic et al. (2016), Chen et al. (2019), Prichard et al. (2019), Jaffe et al. (2020)). Wildfire emissions are dependent on a number of factors such as combustion conditions (e.g., flaming vs. smoldering), fuel conditions (e.g., moisture content), and fuel type (e.g., species and component) (Goode et al. (2000), Urbanski (2013), Liu et al. (2017), Stockwell et al. (2014), Stockwell et al. (2015), Koss et al. (2018), Sekimoto et al.

(2018), Hatch et al. (2019), Prichard et al. (2020)). Differences in these factors can affect the total flux of emissions as well as the profile of emissions, i.e., the identities and quantities of individual constituents. Permar et al. (2021) recently reported that combustion conditions, specifically modified combustion efficiency (MCE), explained approximately 70% of the variability in observed trace gas emissions from wildfires. Consistent with some existing modeling approaches, they suggested total NMOCs could be predicted using MCE, and the contribution of individual compounds determined using speciation profiles. Success of

that approach requires knowledge of the relevant speciation profiles, and therefore contributing fuel types.

NMOC speciation profiles have been developed from both field and laboratory studies (Urbanski et al. (2008), Simpson et al. (2011), Urbanski (2014), Holder et al. (2017), Andreae (2019), and Prichard et al. (2020)). Laboratory studies offer some advantages over field studies in the context of controlling fuel species and fuel components; other variables, such as combustion conditions and fuel moisture, can be harder to control and can lead to differences in the identities and quantities of NMOCs

emitted between laboratory and field studies (Yokelson et al. (2013), Stockwell et al. (2014), Liu et al. (2017), Sekimoto et al. (2018)). Yokelson et al. (2013) presented an inter-comparison of laboratory- and field-based emission factors (EFs), and approaches for using laboratory data to enhance the fundamental understanding of fire emissions coupled with field data to evaluate the representativeness of laboratory-based measurements. At that time, they noted that up to 70% of NMOCs remained unidentified for certain fuel types. More recently, due to the application of advanced instrumental techniques, there has been

significant improvements in the identification and quantification of NMOCs emitted from fires, particularly in laboratory studies (Stockwell et al. (2014), Stockwell et al. (2015), Hatch et al. (2017), Koss et al. (2018)). For example,Stockwell et al. (2015) detected approximately 80–96 % of the total emitted NMOC mass in experiments during the 2012 fourth Fire Lab at Missoula Experiment (FLAME-4); and Hatch et al. (2019) identified more than 500 individual NMOCs during FLAME-4. The relatively rapid expansion in available NMOC data provides opportunities for developing more detailed speciation profiles (in which a

higher fraction of the detected mass is assigned to unique compounds or formulas) and for applying statistical data analysis methods, including to identify unique sets of compounds that allow differentiation of fuel type(s) and estimation of their contributions to smoke samples.





Existing approaches for identifying the contribution of fuel types to smoke include land cover databases or fuel loading models coupled with fuel consumption models (e.g. FOFEM Keane and Lutes (2018) and CONSUME Ottmar (2009)), and

the use of marker compounds. One of the limitations of land cover databases or fuel loading models is that they are difficult to update frequently enough to reflect changes in ecosystems (Reeves et al. (2009), Vogelmann et al. (2011), Nelson et al. (2013) and Lindaas et al. (2021)). Marker compounds are emitted in relatively high abundances and can be used to differentiate fuels by component or fuel layer and in some case by species. For example, Wan et al. (2019) showed that *p*-hydroxybenzoic acid was emitted from combustion of herbaceous plants, while vanillic acid was emitted from combustion of softwoods and

hardwoods. It has also been shown that syringic acid is associated with hardwood combustion (Simoneit (2002) and Zangrando et al. (2013)), and dehydroabietic acid with conifers (Fu et al. (2009)). Zhang et al. (2021) found that the benzene to toluene ratio in smoke from sugarcane leaves was different than the ratio in smoke from sesame stalk, demonstrating differences among agricultural fuels. In measurements of western forests and shrublands, Jen et al. (2018) showed that hydroquinone was a good marker for manzanita combustion. One of the limitations of using marker compounds to identify fuel types is the lack

of specificity, i.e., marker compounds have not been identified that enable identification of a large number of fuel species or closely related fuel species.

In this work a method is presented for identifying fuel types from measured NMOCs in smoke samples. To overcome some of the existing limitations in identifying the contribution of specific fuel types to smoke, pattern recognition (PR) and classification algorithms were developed using data obtained during two laboratory campaigns in 2012 and 2015, and applied

to data obtained during a field study in 2017. Machine learning techniques have been applied for source identification in other disciplines. For example, Welke et al. (2013) and Ziółkowska et al. (2016) used principal component analysis (PCA) and linear discriminant analysis (LDA) to differentiate and classify wine varietals based on specific compounds present in wine samples. Johnson and Synovec (2002) used PCA and analysis of variance to select marker compounds in gasoline fuel blends and pattern recognition (PR) to differentiate the blends. In this work, the large data sets generated during FLAME-4

and the Fire Influence on Regional to Global Environments Experiment (FIREX) 2016 Fire Lab campaign were leveraged to develop a source identification method using fuel-specific NMOC profiles. The PR algorithm performs an automated selection of compounds that differentiate sources (in this case, fuels) based on measured NMOCs. The classification algorithm then uses the source profiles to identify source contributions to specific samples. The data used to train and test the algorithm are introduced in section 2. The algorithm development, implementation, and testing are presented in sections 2 and 3. The

application to field data is presented in section 3; and general conclusions and implications are presented in section 4.

## 2 Data and Methods

### 2.1 Data

The NMOC data used in this study were acquired from a variety of fuel types burned in laboratory and field settings during three campaigns: 1) FLAME-4 laboratory campaign in 2012 (FLAME-4 FL12), 2) FIREX laboratory campaign in 2016 (FIREX

FL16), and 3) Blodgett Forest Research Station (BFRS) prescribed burns in 2017; both laboratory campaigns took place at





the U.S. Forest Service Fire Science Laboratory (FSL). Details of the facilities, sample collection, and data analysis have been discussed in previous publications (Hatch et al. (2015), Hatch et al. (2019) and Hatch et al. (2017)). Briefly, during FLAME-4 FL12 and FIREX FL16 a broad variety of biomass fuels were burned (Stockwell et al. (2014) and Selimovic et al. (2018)), including conifers and shrubs (Table 1); 80 samples were collected from both room and stack burns as described in Stockwell et al. (2014) and Selimovic et al. (2018). During the BFRS study, a total of 28 samples (Hatch et al. (2019)) were collected from a utility task vehicle parked downwind from three different prescribed burn plots that had different fuel distributions (see Supplementary Information (SI) Fig. S2-S4 in Hatch et al. (2019) and Table 1). All NMOC samples were collected using dual bed stainless steel sorbent tubes and were analyzed using an automated thermal desorption unit coupled to a two dimensional gas chromatograph with a time-of-flight mass spectrometer (GC × GC-TOFMS). The raw chromatograms were processed using the commercially available software Chromatof (Leco Corp., St. Joseph, MI). The measured mixing ratios were used to calculate normalized excess mixing ratios (NEMR) versus CO, $\Delta X/\Delta CO$ (Yokelson et al. (1999)), in which delta represents excess over background. The calculated NEMRs of monoterpenoids ($C_{10}H_{16}$ and $C_{10}H_{16}O$) were used as the starting point for this analysis based on Hatch et al. (2018) and Hatch et al. (2019). Hatch et al. (2019) demonstrated that the variability in NMOC composition could not be attributed entirely to MCE, and that chemical speciation was highly correlated among some fuel types across a range of MCE values, particularly conifers; within conifers, clear differences in monoterpenoid emissions were observed as a function of fuel species.

## 2.2 Pattern recognition algorithm

A four-step PR algorithm (Fig. 1) was developed to select a subset of compounds that captured the variance between the fuel types and then use those marker compounds to differentiate fuel types based on the selected NMOCs; the algorithm steps are: 1) data preprocessing, 2) feature selection, 3) principal component analysis, 4) and k-means clustering. The algorithm was implemented using the Python package scikit-learn (Pedregosa et al. (2011)). The algorithm steps are explained in more detail in the following sections.

### 2.2.1 Preprocessing and feature selection

Data preprocessing (step 1) is performed to handle any missing values in the samples, largely as a result of compounds being below the detection limit. During preprocessing, for every feature (i.e., compound) the percentage of missing values across all samples is calculated. For any given compound, if the percentage by number of missing values is less than 30% (Schafer (1997)) then the missing values are replaced with zeros. If the percentage by number of missing values is more than 30% then the compound is removed from the dataset. Feature selection (step 2) reduces the number of variables by selecting only informative ones. In this work an analysis of variance (ANOVA)-based feature selection method, similar to Johnson and Synovec (2002) and Welke et al. (2013), was used to further filter all compounds retained in step 1. Finally the selected compounds were standardized (Lever et al. (2017)) before applying PCA, since PCA is sensitive to variables with different scales, which can bias the output.



**Table 1.** Fuels Burned and Smoke Analyzed From FLAME-4 2012 Laboratory Fires, FIREX 2016 Laboratory Fires, and Blodgett Forest Research Station Prescribed Burns.

| Fuel Family | Fuel Type | FLAME-4 FL12 (lab) | FIREX FL16 (lab) | BFRS |
|---|---|---|---|---|
| *Conifers* | | | | |
| | Ponderosa pine | x | x | x |
| | Lodgepole pine | | x | |
| | Engelmann spruce | | x | |
| | Black spruce | x | | |
| | Douglas fir | | x | |
| | Subalpine fir | x | x | |
| | White fir | | | x |
| | Juniper | | x | |
| | Loblolly pine | | x | |
| | Sugar pine | | | x |
| | Jeffrey pine | | x | |
| | Incense cedar | | | x |
| *Shrubs* | | | | |
| | Chamise | | x | |
| | Manzanita | | x | |
| | Sagebrush | | x | |
| | Snowbrush ceanothus | | x | |
| *Miscellaneous* | | | | |
| | California black oak | x | | |
| | Tanoak | | | x |
| | Excelsior | | x | |
| | Yak dung | | x | |
| | Peat | x | x | |
| | Rice straw | x | x | |
| | Bear grass | | x | |
| | Untreated lumber | x | | |

### 2.2.2 Principal component analysis and k-means clustering

PCA (step 3), as described in Abdi and Williams (2010), is a dimensionality reduction technique that is used to project high dimensional data into a lower dimensional space along the direction(s) of maximum variance in the data. k-means (Jain (2010)) (step 4) is a popular clustering algorithm that finds clusters in a n-dimensional space (Jolliffe (2002) and Abdi and Williams





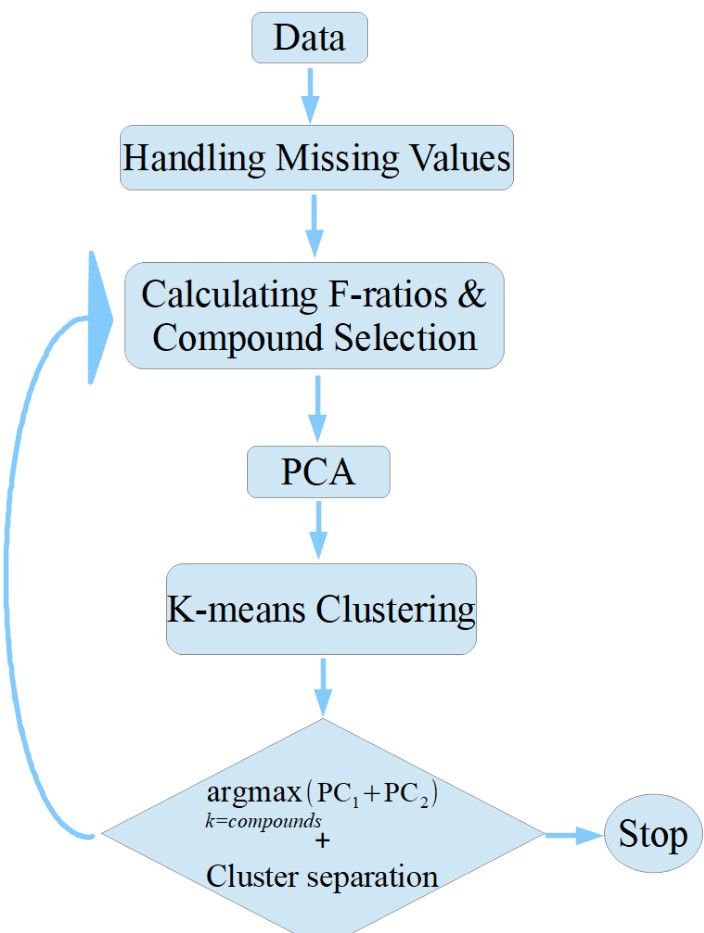

**Figure 1.** Pattern recognition algorithm flowchart

(2010)). In this study PCA was used to compress the information carried by the selected compounds to a lower dimensional space and k-means clustering was used to find formed clusters after the application of PCA.

## 2.3 Classification

To test the applicability of the PR algorithm results for field samples, a classification algorithm, LDA (Hastie et al. (2009)), was applied. LDA is a supervised learning method that is similar to PCA. Both LDA and PCA are linear transformation techniques: LDA is supervised, whereas PCA is unsupervised and ignores class labels. While PCA tries to find a subspace of features in order to maximize variance among samples, LDA attempts to find a feature subspace that maximizes class separability. The input for the LDA training were the selected principal components from the pattern recognition analysis as the independent





**Table 2.** Data sets used for developing pattern recognition algorithm and testing and training classification algorithm.

| Data Set | Pattern Recognition | Training Set | Testing Set |
|---|---|---|---|
| FLAME-4 FL12 | | | x |
| BFRS | | | x |
| FIREX FL16 | x | x | |
| Synthetic data | x | | x |

variables and the fuel types as the response variable/class (see section 1 in SI). The output of LDA is a probability score for every sample that is being tested for its likelihood to belong to a particular fuel type.

## 3 Algorithm Implementation, Results and Discussion

### 3.1 Sample and fuel type selection for pattern recognition and classification

The PR algorithm was applied to the FIREX FL16 data set to identify a group of marker compounds that could be used to
differentiate fuel types. Classification was then performed using the FIREX FL16 data as the training set, and BFRS data as the testing set. The selection of the training and testing sets was based on the size of each data set; the FIREX FL16 data set had 74 samples, and the BFRS data set had 29 samples. A larger training set ensured more statistically robust parameters for the LDA algorithm. Because the BFRS data span a wide range of complexity in the fuels sampled, a synthetic data set was generated to test the performance of the PR and classification algorithms on mixed fuel samples prior to application on the BFRS data.
Five synthetic mixtures were generated with the following compositions: 60% pine/40% spruce, 60% fir/40% spruce, 60% pine/40% fir, 90% pine/10% spruce, and 90% fir/10% spruce. The FLAME-4 FL12 data were used as an independent data set to test the response of the classification algorithm to fuel types that were not included in the training set. The use of each data set in the PR and classification algorithms is summarized in Table 2.

Two selection criteria were applied to the training set to ensure that standard deviations and averages could be computed,
which are central features of the PR algorithm. First, only fuel types that had more than 30% (by number) of the 93 monoterpenoids above the limit of detection (LOD) were selected. Since the PR algorithm was based on monoterpenoids, samples with little to no detected monoterpenoids would reduce the ability of the algorithm to differentiate between fuel types and therefore reduce the overall efficiency. Second, only fuel types that had three or more samples were retained. Application of these criteria reduced the number of samples from a total of 74 to 39 and the number of fuel species from 18 to five: pines (ponderosa pine
and lodgepole pine), firs (Douglas fir and subalpine fir) and spruce (Engelmann spruce). During the FIREX FL16 study different fuel components were also burned such as canopy, rotten log, composite, litter and duff. While differences in component emissions may be important for differentiating prescribed burn and wildfire emissions in smoke, for this application,based on the selection criteria, 32 composite and canopy samples were retained along with seven litter and duff samples.





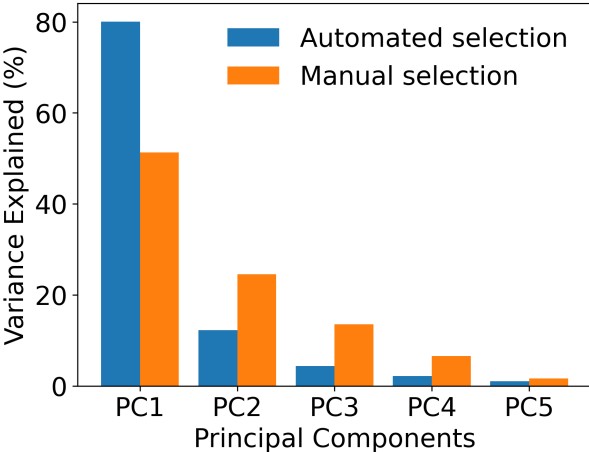

**Figure 2.** Percentage of variance explained per principal component (PC) for the case of automated (blue) and manual (orange) compound selection using Fisher ratios.

## 3.2 Pattern recognition

### 3.2.1 Feature selection

Feature selection was performed and evaluated using two approaches: 1) manual selection; and 2) automated selection based on Fisher ratios (F-ratios). The percentage of variance explained using the first two principal components (PCs) was used as the metric to evaluate the quality of feature selection using the two approaches (see scree plot method section 3.2.2) following application of PCA (Fig. 2). For manual selection, the compounds were filtered based on a single criterion: whether a compound

was present in more than three fuel species. For automated selection, the F-ratios were calculated for every compound in the data set using Eq. 1. In Eq. 1 the nominator ($V_b$) corresponds to the between-class sum of squares and the denominator ($V_w$) to the within-class sum of squares.

$$F_{ratio} = \frac{V_b}{V_w} (1)$$

The F-ratio provides an indication of class separability, in which higher ratios indicate higher discriminatory power between

classes. In this application samples were separated into classes based on the fuel family (firs, pines and spruce). Following the F-ratio calculation the compounds were ranked in ascending order based on their F-ratio values. Then, in an iterative fashion, PCA was performed and the explained variance from the first two PCs (Fig. S1) and the cluster separation were evaluated as a function of the number of compounds (with the highest F-ratios) retained.

The application of the manual approach resulted in the selection of the following nine compounds (out of 93): a-pinene,

limonene, 3-carene, b-myrcene, camphene, p-cymene, bornyl acetate, b-phellandrene, and tricyclene. Application of the auto-





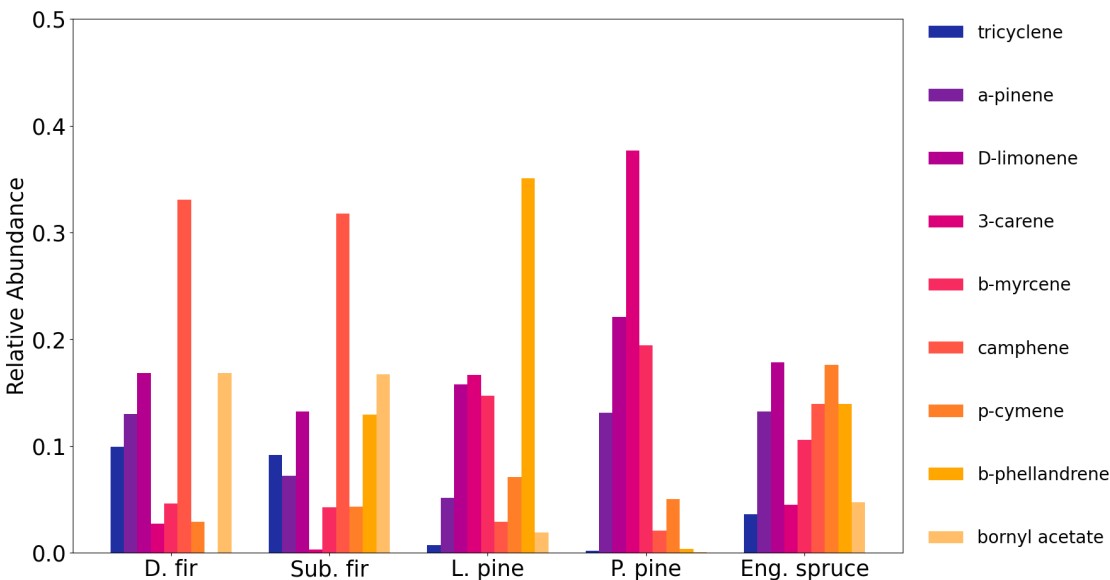

**Figure 3.** Normalized emission ratio profiles for: Douglas fir, subalpine fir, lodgepole pine, ponderosa pine, and Engelmann spruce based on manual selection of compounds.

mated approach resulted in selection of the following five compounds: tricyclene, camphene, b-pinene, 3-carene, and bornyl acetate. Figure 2 shows the improved performance of automated feature selection over manual feature selection, based on the single highest explained variance across PCs 1-4. To make the feature selection results more intuitive, the normalized emission ratio profiles (ratio of the compound ER to the sum of ERs for the selected compounds) as a function of fuel species are shown for manual selection (Fig. 3), automated selection for five compounds (Fig. 4) and ten compounds (Fig. S2). The automated selection with five compounds results in more distinct and consistent profiles for each fuel family, which translates to a higher potential for separation (greater explained variance) in the PCA space. The five compounds selected with the automated approach were thus used for the PR analysis.

### 3.2.2 PCA and k-means clustering

Following data preprocessing and feature selection, PCA was performed on the reduced data set. To determine the number of PCs to be retained, a scree test using a modified version of the Kaiser criterion (Jolliffe (2002)) was performed. In the scree test, the component number is determined by plotting the acquired eigenvalue (or explained variance) as a function of component number. A steep decrease or inflection point indicates the number of usable components. In this study the normalized eigenvalues were calculated using Eq. 2 and plotted against the number of components (Fig. 2).

$$\frac{I_j}{\sum_{j=1}^{p} I_j} (2)$$




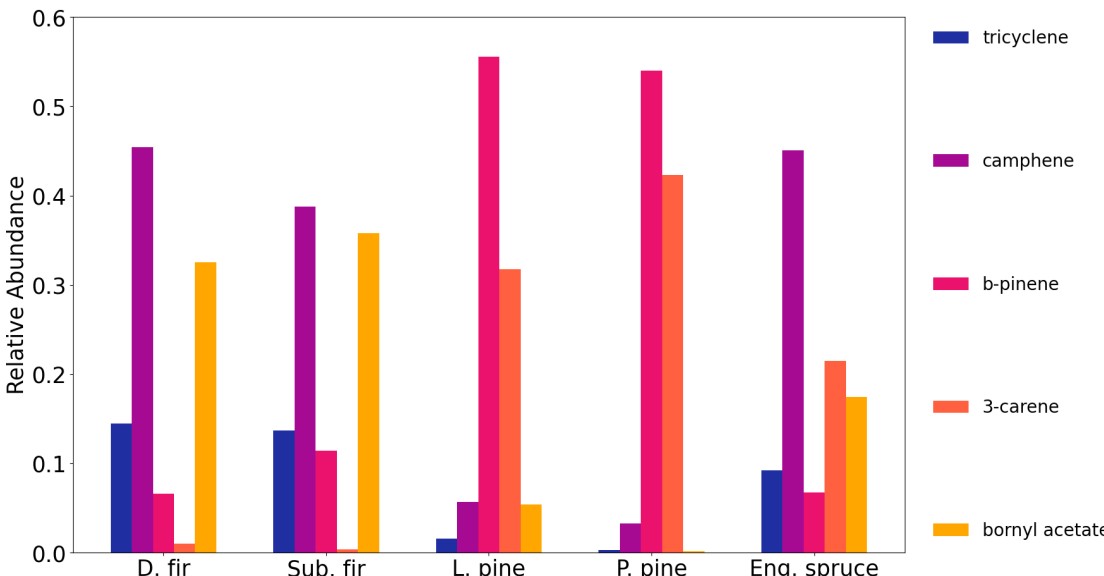

**Figure 4.** Normalized emission ratio profiles for: Douglas fir, subalpine fir, lodgepole pine, ponderosa pine, and Engelmann spruce based on automated selection of compounds.

where $I_j$ corresponds to the eigenvalue for PC$_j$. Figure 2 shows that with automated feature selection two PCs (PC1 and PC2) were adequate to explain 92% of the variance in the data set. The scores from the retained PCs (PC1 and PC2) were then used as input for k-means clustering. To determine the optimal number of clusters, the elbow plot method was used. The elbow plot is a graphical method in which the cumulative distance is calculated for all points/samples from their respective centroids and

then plotted against the number of clusters. In this study Euclidean distance was used (Eq. 3):

$$d = \sum_{k=1}^{k_{max}} \sum_{j=1}^{n} \sqrt{(q_j - c_k)^2} \, (3)$$

where $q$ and $c$ are multidimensional vectors with the coordinates for each centroid and sample, respectively. The values for each sample are the respective scores from the selected PCs. The indices $j$ and $k$ correspond to centroid number and sample number, respectively. The optimum number of clusters is found after a steep decrease in the total Euclidian distance, followed

by trivial changes, with increasing number of clusters. A steep decrease signifies good clustering performance (dense clusters), while a trivial change in the Euclidian distance shows that the cluster centroids do not change substantially from their previous positions. The elbow plot (Fig. 5) shows that the algorithm identified four clusters as the optimum number.

     The coupled PCA and k-means results are shown in Fig. 6. The clusters identified by the algorithm are differentiated using marker color while the fuel families are differentiated using marker shape. Cluster one included 15 out of 16 pine samples and

one overlapping fir sample. Cluster two included 11 out of 13 fir samples and four overlapping spruce samples. Clusters three and four included the remaining six spruce samples, one overlapping pine sample, and one overlapping fir sample. Generally

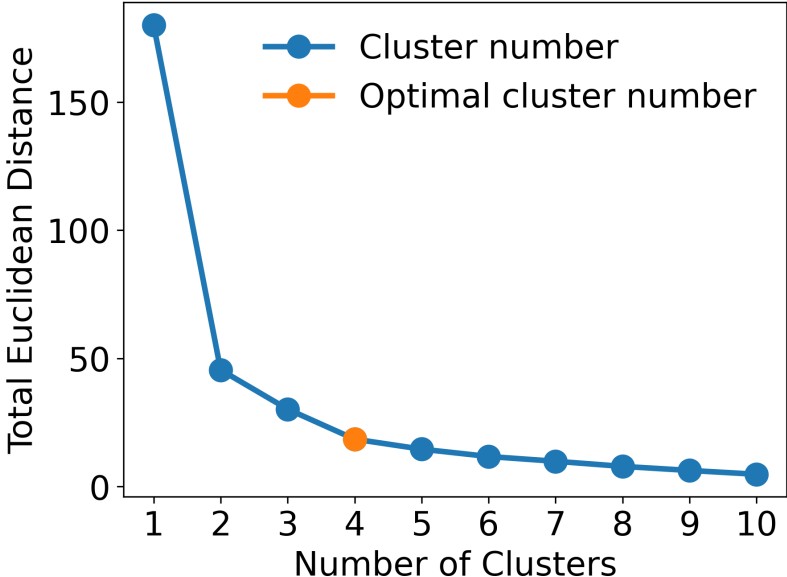

**Figure 5.** Elbow plot for k-means clustering with automated compound selection for the PC1 and PC2 pair. The orange marker indicates the optimum number of clusters.

the algorithm resulted in adequate separation between firs and pines; but poorer separation for spruce, for which four of ten samples overlapped with another fuel family. Adding more compounds (10) reduced the explained variance from PC1 and PC2 from 92% to less than 70% (Fig. S1) and resulted in only minor improvement in cluster separation (Fig. S3). The difficulty that

the algorithm encounters separating spruce (Fig. 6) effectively can be explained using the elbow plot (Fig. 5). The k-means algorithm identified four clusters as the optimum number, but the steep decrease in the total Euclidean distance actually occurs between one and two total clusters. The Euclidean distance decreases more between two and four but to a lesser extent (smaller slope) compared to the decrease between one and two. The lesser decrease between two and four clusters indicates that the clustering algorithm had difficulty identifying clusters in the PCA space, which is then apparent in Fig. 6. From the normalized

emission ratio profiles (Fig. 4), it can be seen that the spruce and fir samples have similar normalized tricyclene, camphene, and b-pinene emission ratios. This limits the ability to fully separate spruce and firs in the PCA space.

### 3.2.3 Beyond principal components one and two

Thus far, only the first two PCs (from a total of five) were used in the analysis since they explained 92% of the variance in the data set. Another 8% is shared between PCs three and four, which could potentially provide better separation for spruce

samples. After testing PCs one with three and one with four, the combination of one and four resulted in better performance of the PR algorithm. Even though the optimal number of clusters for the PC1 and PC4 pair (Fig. 7) is the same as with the PC1 and PC2 pair it was found at a lower total Euclidean distance, which is indicative of more dense clustering. Figure 8





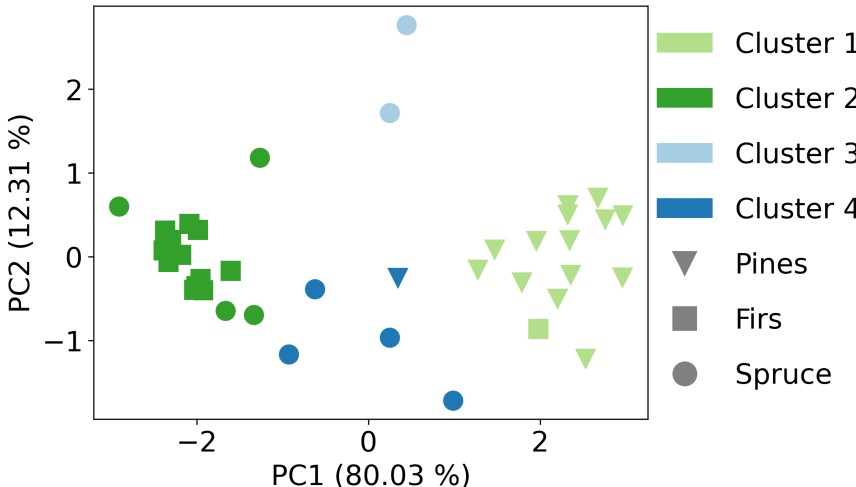

**Figure 6.** PCA coupled with k-means clustering results for the PC1 and PC2 pair.

shows the results for the PC1 and PC4 pair. Cluster one included 13 out of 16 pine samples and one overlapping fir sample. Cluster two included 11 out of 13 fir samples and only one overlapping spruce sample. Clusters three and four included nine

spruce samples out of 10, one overlapping fir sample and three overlapping pine samples. With the PC1 and PC4 pair, spruce samples have 30% less overlap with firs (Fig. 9), with only moderate losses in the separation between spruce and pines. These results demonstrate the ability of the PR algorithm to separate firs, pines, and spruce in the smoke samples, with only two PCs accounting for most of the variance in the data set (PC1 and PC4 about 82%).

### 3.2.4 Mixed samples

The PR algorithm selected compounds that separated smoke samples by the contribution of three individual fuel families (firs, pines and spruce). To test the algorithm for mixed fuel samples, as would be more common in the field, five synthetic fuel mixtures were used: 60% pine / 40% spruce, 60% fir / 40% spruce, 60% pine / 40% fir, 90% pine / 10% spruce, and 90% fir / 10% pine. From the three 60/40 samples only the fir/spruce synthetic mixture was clustered with the dominant fuel family (fir). The pine/spruce and pine/fir synthetic mixtures were clustered with spruce (Fig. 10 ), clusters one and three respectively.

The grouping of the pine/spruce synthetic mixture as spruce is marginal in the PCA space and is due to the scatter of the spruce samples rather than the similarity of the synthetic mixture with spruce. Of the three 60/40 synthetic mixtures, the pine/fir mixture was grouped with the spruce clusters. The grouping of the pine/fir mixture with spruce is more intuitive after comparing the normalized ER profile with that of spruce (Fig. S4), where the ER profile for the pine/fir mixtures is similar to spruce. Figure 11 shows the PR results including the 90/10 synthetic mixtures. Both samples were grouped correctly with their

respective dominant fuel family.




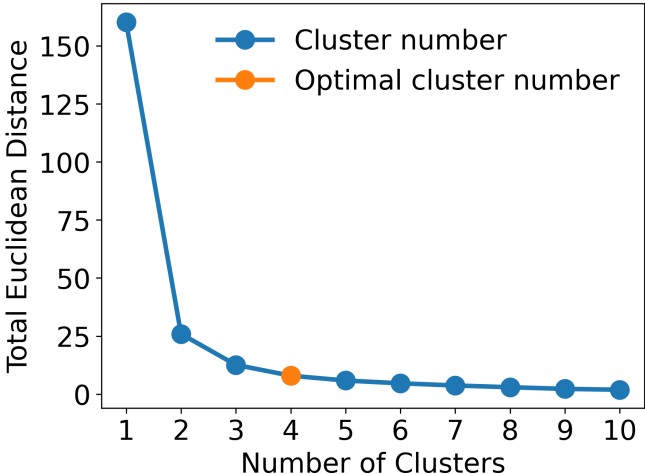

**Figure 7.** Elbow plot for k-means clustering with automated compound selection for the PC1 and PC4 pair. The orange marker indicates the optimum number of clusters.

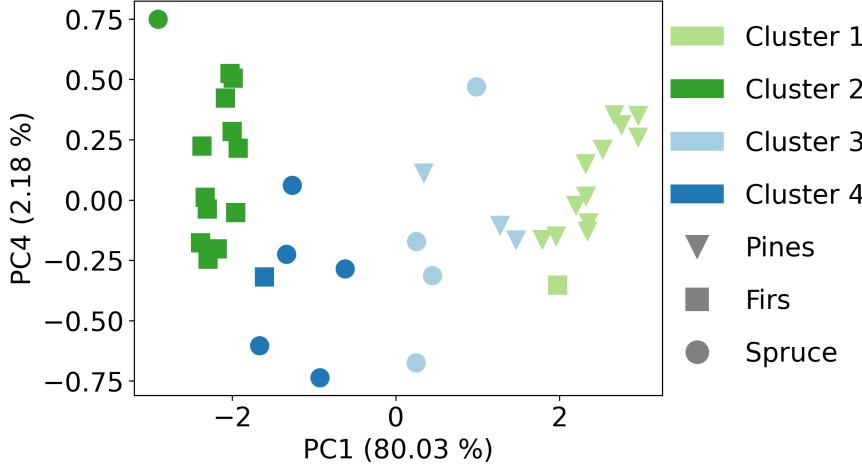

**Figure 8.** PCA coupled with k-means clustering results for the PC1 and PC4 pair.

The results with the synthetic mixtures suggest that the algorithm can select marker compounds that can differentiate fuel types even when they are highly mixed (60/40 cases for pine/spruce and fir/spruce) but for some mixtures (60/40 pine/fir) the differentiation might be poor. One approach for improving separation using the PR algorithm, could be to incorporate more mixed fuel samples in the training and test sets.



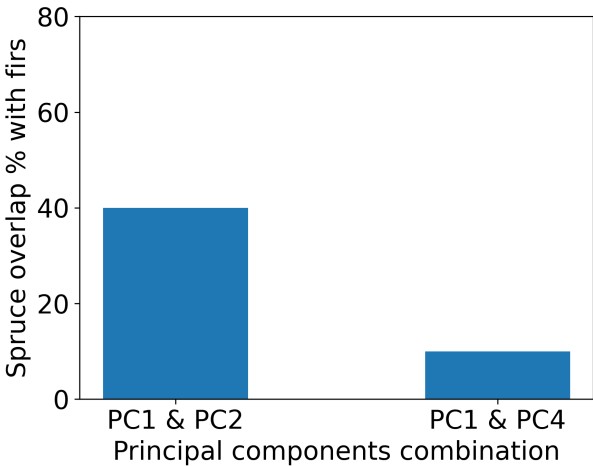

**Figure 9.** Spruce overlap with firs based on the PC combination used.

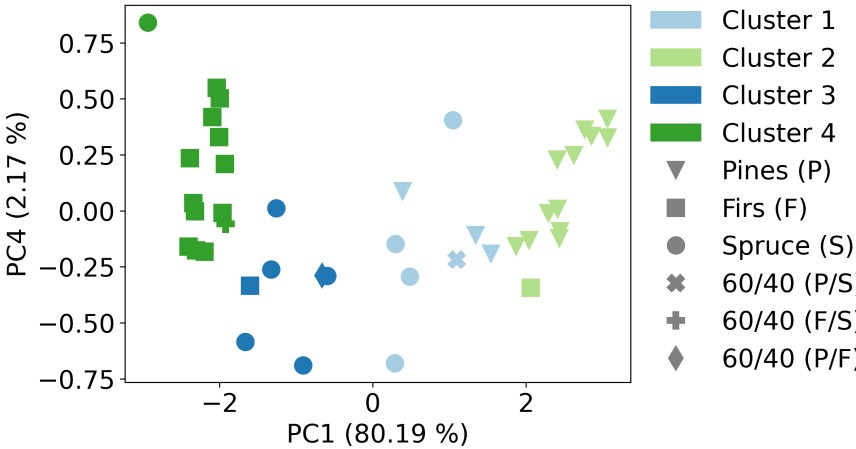

**Figure 10.** PCA coupled with k-means clustering results for the PC1 and PC4 pair including the 60%/40% synthetic mixtures.

## 3.3   Classification

For the classification algorithm, the scores of the selected PCs were used as input for the LDA training. PC1 and PC4 were selected since they provided better separation across the three fuel families (see section 3.2.3), while explaining 82% of the variability in the data set. The outcome of LDA is a probability, calculated using Eq. 4, that a sample belongs to one of the main clusters determined by k-means (see section 3.2.3).

$$\log P(y=k|x) = -\frac{1}{2}(x-\mu_k)^t \sum{}^{-1}(x-\mu_k) + logP(y=k) + C_{st}\ (4)$$





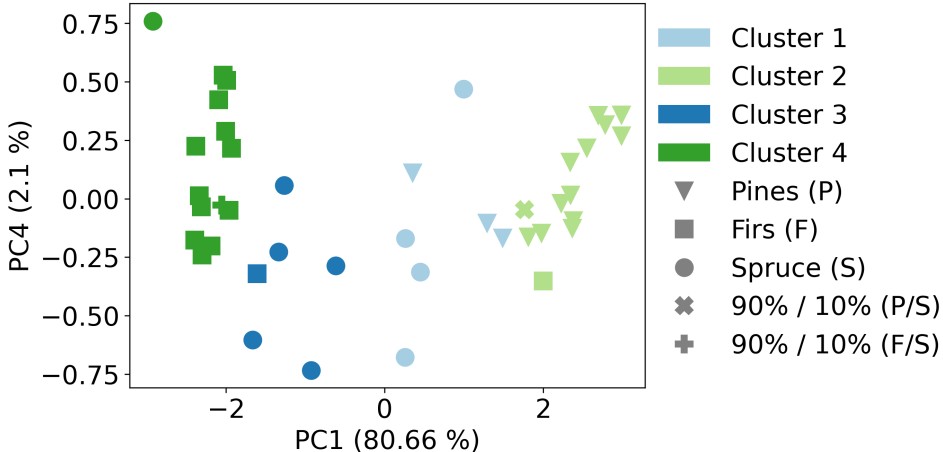

**Figure 11.** PCA coupled with k-means clustering results for the PC1 and PC4 pair including the 90%/10% synthetic mixtures.

where $k$ is the class of sample $x$, $\mu$ is the vector of the means for each class based on the selected features and $\Sigma$ is the common covariance matrix for the three classes in the training set. In this application the probability score is related to the proximity of sample to a class of samples (cluster) in the PCA space (Fig. 10 and Fig. 11) which is linked to its similarity with the emission profiles for the three fuel families (Fig. 4). The assignment of a sample to a class is based on the class with the highest

probability, even if marginally higher. For example a sample with a pine probability score of 70% or more will most likely be inside the pine cluster. Generally, samples with probability scores 60% and higher are most likely in the cluster space of a fuel family. Samples with a probability score 60% and lower are more likely to be adjacent to more than one fuel family in the PCA space. For the training of the classifier all 39 samples from FIREX FL16 were used.

### 3.3.1   Synthetic mixtures and FLAME-4 FL12 samples

The classification algorithm was tested using the synthetic mixtures and FLAME-4 FL12 samples before testing using the BFRS field data. The classification results for the synthetic mixtures are shown in Fig. 12. Two of three 60/40 synthetic mixtures were classified correctly, pine/spruce and fir/spruce, with classification probabilities of 70% and higher for the dominant fuel family. The 60/40 pine/fir synthetic mixture was classified as spruce. Its classification is a result of its clustering in the PCA space with spruce (Fig. 10), which is directly connected to its similarity with the spruce emissions profile (Fig. S4). The two

90/10 synthetic mixtures, pine/spruce and fir/spruce, were correctly classified with classification probabilities over 80% for the dominant fuel family. Application of the classifier to the synthetic mixtures demonstrated that the mixtures can be correctly classified based on the dominant fuel family in mixed fuel samples (4 of 5 mixtures); however, incorrect classification can occur when the mixed fuel emissions profiles are similar to individual fuel emissions profiles, resulting in poorer separation with PCA. The results of the classification algorithm for mixed samples can be improved in future work through expanded

testing and training on a broader range of fuel families and relevant mixtures.





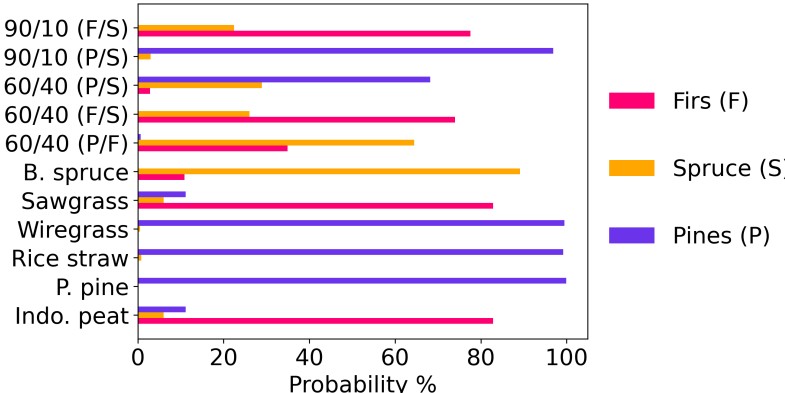

**Figure 12.** Classification results for synthetic mixtures and FLAME-4 samples.

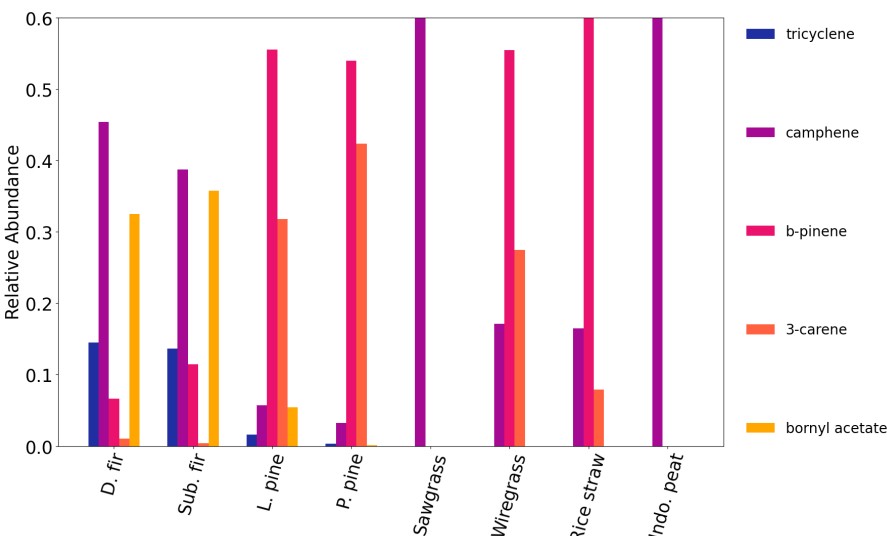

**Figure 13.** Normalized emission ratio profiles for FIREX FL16 samples: pines and firs; and FLAME-4 FL12 samples: sawgrass, wiregrass, rice straw, and Indonesian peat. The relative abundances of camphene and b-pinene in sawgrass, Indonesian peat, and rice straw were >0.6, but for figure clarity, the axis limits were not changed.

The classification results for the FLAME-4 FL12 samples are shown in Fig. 12. This data set included six fuel species (ponderosa pine, black spruce, Indonesian peat, rice straw, wiregrass, and sawgrass); only one of which, ponderosa pine, was in the training set. Both ponderosa pine and black spruce samples were classified correctly (Fig. 12), with classification probabilities over 90%. The Indonesian peat, rice straw, wiregrass, and sawgrass samples were classified as firs or pines (Fig. 12), with classification probabilities over 70%. The classification algorithm evaluated partial similarity against only three options (pine, firs or spruce), none of which represent the fuel families of the four fuel species. Figure 13 shows the average





normalized emission ratio profiles for pines and firs, as well as Indonesian peat, rice straw, wiregrass, and sawgrass. It can be seen that only camphene appears in the sawgrass and Indonesian peat samples, and thus these fuels are classified as firs, which also have a high relative abundance of camphene. Wiregrass and rice straw samples also include camphene, but have higher
relative abundances of b-pinene and 3-carene, and thus were classified as pines, which also have higher relative abundances of these two compounds (Fig. 13). As illustrated by the application to the synthetic fuel mixtures, the performance of the classification algorithm can be improved in future work by expanding the range of fuel families and mixtures included in the training and test sets.

### 3.3.2 Blodgett samples

Figures 14-16 show the results of the classification algorithm applied to the BFRS samples from three different prescribed burn plots: 60, 340, and 400. Based on the fuel bed composition (see SI Fig. S2-S4 in Hatch et al. (2019)) there was a total of seven different fuel species in the three burned plots: white fir, incense cedar, tanoak, sugar pine, ponderosa pine, Douglas fir, and California black oak. Due to the heterogeneity of the fuels, and the influence of meteorology and sampling location, it was not possible to determine the relative contribution of each fuel species to each sample. Instead the average overstory composition
(Figs. S5-S8) was used to determine likely influences from dominant sources close to each sampling location. For plot 60 (sites one and two) (Fig. S5) the main influence was from firs (47%) followed by similar amounts of pines and incense cedar (25% and 27%) with no contribution from tanoak or California black oak. For site three in plot 60 (Fig. S6) the main influence was from incense cedar (43%) followed by firs (34%), pines (26%), and California black oak (10%). For plot 340 (Fig. S7) the main influence was from firs (63%) followed by pines (21%), incense cedar (12%), and (2%) from tanoak and California black oak.
Finally for plot 400 (Fig. S8) the main influence was from firs (55%) followed by pines (26%) and incense cedar (18%). The classification algorithm classified all samples from plots 60 (Fig. 14) and 340 (Fig. 15) as fir dominant. Nine out of ten samples from plot 400 (Fig. 16) were classified as fir dominant and one as pine dominant. While spruce was absent in the burned plots, all samples (with the exception of the pine dominant sample in plot 400) had a higher classification probability for spruce than pines.

For plot 60 there were a total of 11 samples collected. Five samples in sites one and two, which is fir dominant (Fig. S5) and six samples in site three which is incense cedar dominant (Fig. S6). For sites one and two the classifier results are reasonable based on the overstory composition, but for site three the classification results are inconclusive since no emissions profiles were available for incense cedar which does not belong to the fir fuel family. It is likely that incense cedar or the mixture of incense cedar with firs most closely resembles the fir emissions profile of the selected compounds and thus was classified
as firs. For plots 340 and 400 the classification results are reasonable, 16 out of 17 samples (Figs. 15-16) are fir dominant, since in both plots the influence of firs is above 50% with 63% contribution in 340 and 55% in 400. The one sample that was classified as pines in plot 400 was most likely affected strongly by ponderosa pine emissions during sampling. SI Fig. S4 (plot 400) in Hatch et al. (2019) shows that one of the inventoried land plots next to sites one and two had an average fractional overstory composition of ponderosa pine of more than 50%. Regarding the elevated probability for spruce, despite its absence
from all burned plots, it is likely an artifact of mixed smoke between pines and firs. In section 3.3.1 the analysis showed that





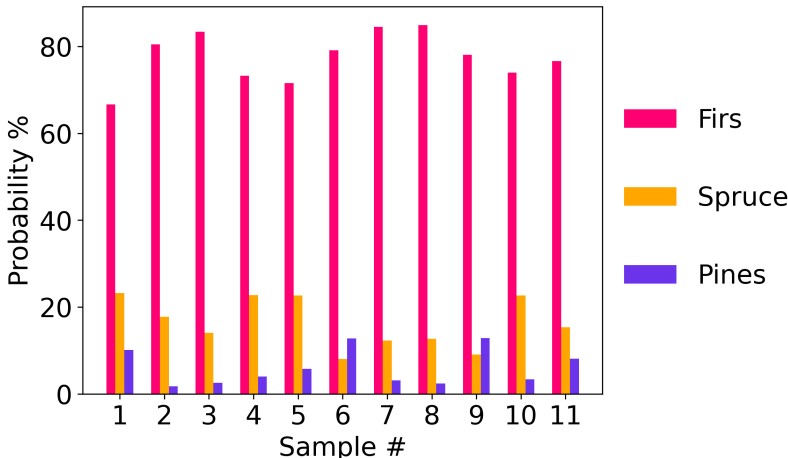

**Figure 14.** Classification probability by fuel class for plot 60.

the synthetic mixture between pine and fir was more similar to spruce (Fig. 10 and Fig. S4). Among the three plots, pines and firs together account for more than 70% on average of the overstory composition. Thus the contribution from both firs and pines could lead to smoke mixtures that resemble the spruce emissions profile. Tanoak and California black oak (Figs. S5-S8) account for 2% - 10% of the total contributions among the three plots. Due to insufficient data regarding their emission profiles

of the selected compounds their true contribution to the smoke samples could not be evaluated, but given their low overstory contribution it is likely that they did not influence the collected samples substantially. The results for the BFRS data showed that the lab-based emission profiles selected by the PR algorithm, can be applied to smoke samples collected in the field and can detect dominant fuel sources even in mixed smoke samples. While the algorithm has been tested and trained on only three fuel families, widespread application can be achieved with further training and testing using a more diverse set of compounds

and broader range of fuel types.

## 4 Conclusions

### 4.1 Pattern recognition and classification

A supervised pattern recognition (PR) algorithm was developed and applied in this study to: 1) differentiate sources/fuel types using NMOCs measured in smoke samples and selected with an ANOVA based feature selection method; and 2) train

a classification algorithm to detect dominant sources/fuel types in smoke samples based on the unique speciation profiles identified by the PR algorithm. The PR algorithm was able to group five fuel species (Douglas and subalpine fir, ponderosa and loblolly pine, and Engelmann spruce) into three fuel families (pines, firs and spruce), with minimum overlap; only 5 of 39 total samples were grouped with families that were not representative of the fuel species. The separation was achieved using five





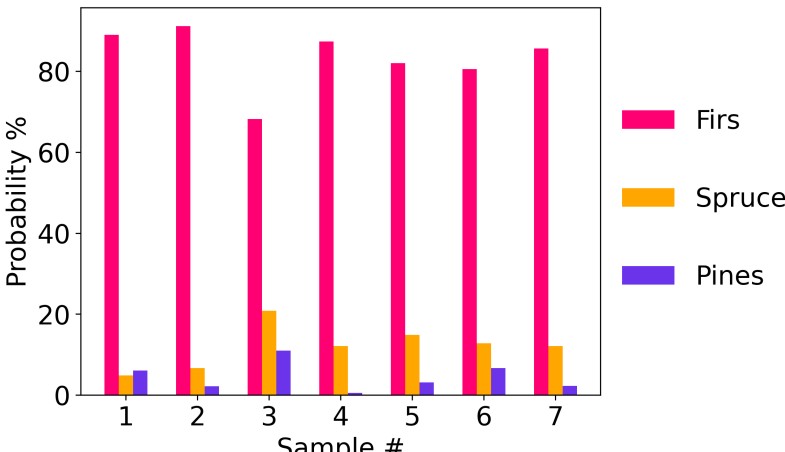

**Figure 15.** Classification probability by fuel class for plot 340.

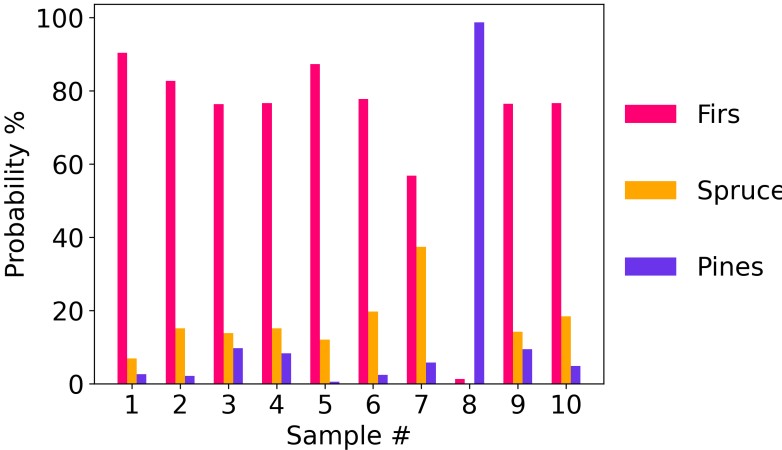

**Figure 16.** Classification probability by fuel class for plot 400.

monoterpenoids that the algorithm selected out of a pool of 93. The PR algorithm was tested with five synthetic fuel mixtures
where it successfully separated three of five (60/40 fir/spruce, 90/10 pine/spruce, and 90/10 fir/spruce) and grouped them with
their dominant fuel type. The same synthetic fuel mixtures were also tested using the classification algorithm, where four of
five were classified correctly (60/40 pine/spruce, 60/40 fir/spruce, 90/10 pine/spruce, and 90/10 fir/spruce). The application of
the classification algorithm to the synthetic mixtures demonstrated that dominant source contributions could be identified in
fuel mixtures. For the FLAME-4 FL12 samples the classification algorithm correctly classified two of six samples (ponderosa
pine and black spruce); these two samples were the only fuels represented by the three fuel families. For the BFRS field



samples, based on the fractional overstory composition, the classification results were reasonable with 27 out 28 samples being classified as fir dominant and one sample as pine dominant. The incorrect classifications that occurred with the synthetic fuel mixture (60/40 pine/fir) and the FLAME-4 FL12 samples (Indonesian peat, rice straw, wiregrass, and sawgrass) were due to the similarity or partial similarity of their emissions profiles with the fuels used to train the classification model. This can be resolved in future applications by including more compounds and a broader range of fuel types, including in mixtures. This will also facilitate the use of this approach for identifying contributing fuels outside of Western coniferous forests.

*Code and data availability.* The data and implementation scripts for the pattern recognition algorithm and the classification model are available through a github repository https://github.com/christos-stamatis/supervised_pattern_recognition but are not yet final.

*Author contributions.* CS and KCB contributed equally to the manuscript; CS lead the data analysis, data interpretation, and manuscript preparation efforts; KCB lead the conceptual design and manuscript editing.

*Competing interests.* The authors declare that they have no conflict of interest.

*Acknowledgements.* The authors want to thank; Lindsay Hatch for data acquisition and processing of all FIREX FL16, FLAME-4 FL12, and BFRS samples; Ariel Roughton for the land cover analysis of the BFRS samples; Paul Van Rooy for discussions on the applicability of speciation profiles for data interpretation and smoke modeling; and Vanessa Selimovic for the CO data that were used for the ER calculations.

*Financial support.* This work was supported by; NOAA grants NA16OAR4310103 and NA17OAR4310007; and CARB grant 00010311.





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
