# Peer review of "Development and Application of a Supervised Pattern Recognition Algorithm for Identification of Fuel-Specific Emissions Profiles"

_Atmospheric Measurement Techniques, 2021_

## Referee Comment (RC1)

Review of Stamatis et al: "Development and Application of a Supervised Pattern Recognition Algorithm for Identification of Fuel-Specific Emissions Profiles"

**General comment**

The manuscript introduces interesting algorithm for identification of different fuel types from emission profiles. The algorithm is a combination of widely applied statistical methods which together make a good tool for emission classification. However, the methods need to be described with more detail and the manuscript needs to be structured better. The methods should be separated from the results, now the text is difficult to read as new methods pop out from nowhere in the middle of results. With these and the detailed comments below properly addressed I could recommend the manuscript for publication.

**Detailed comments:**

Page 4 lines 115-118: IS the reason for missingness always the compounds being below the detection limit? If there are missingness due to e.g. instrument malfunction, filling in the zeros might bias the further analysis.

Page 4, lines 119-122: Feature selection needs a bit more clarification. ANOVA is sensitive to non-normality and heteroscedasticity of the data and if this is not taken account the feature selection may be biased. Whereas for PCA, standardization should not be standard procedure (Gewers et al. 2018), especially if the components are calculated with singular-value decomposition (SVD, Isokääntä et al. 2020) as I assume was applied here since the authors refer to Abdi&Williams 2010 paper. In addition, it should be stated in the methods section if SVD or eigenvalue decomposition was used to find the principal components.

The whole section introducing the PR algorithm should be extended to give more details on the methods used. It should be stated why PCA was chosen over to other dimension reduction methods, like explorative factor analysis, and similarly for LDA. PCA execution should be described with more details. In addition to decomposition method, it should be stated if some rotation method was used and which type of rotation: orthogonal or oblique and which version of them. Rotation methods are discussed in Abdi&Williams and in Isokääntä et al. (2020).

Section 3.2.1. The feature selection needs more justification. As PCA is, by definition, dimension reduction method, why the dimensions need to be reduced before running PCA for PR? Was the explained variance used as selection criteria for lower number of compounds in the analysis? It should not be as higher number of variables means higher total variance, which in turn leads to lower explanation rate with same number of components. Is the result in Fig 2. for the selected 5 (or 10) variables? I yes, then it should be noted that with five components you will explain 100% of the variation of five variables (but not for 10 variables). Since PCA tries to explain most of the variance with the first component, it is expected that the explanation rate is high. Naturally, with statistical models the aim often is to get as simple model as possible but the selection criteria need to be well reasoned.

Line 189. Fig 2 does not show normalized eigenvalues. Add the scree plot.

Line 195: Why Euclidean distance? I am not questioning the choice but wish to see the reasoning

Section 3.2.3: Did I understand correctly that you were only using one pair of components at the time in cluster analysis? If yes, I would ask why? Cluster analysis is a multivariable method, and thus can be applied even for all components at the same time. Figure 8 shows that PC4 does not really affect the clustering, but it is defined by PC1. Thus, this paired comparison seems redundant.

Page 13, line 243-244: Separation would be improved also by increasing the number of compounds in the analysis or using different rotation in PCA.

Results and Conclusions: Too strict limitations in feature selection also reflects to results and conclusions. Using more features in the model would probably lead better separation of sources in mixed samples but in creasing the number too much would decrease generalizability of the model or even lead to overfitting. Thus, it is important to find the balance.

**References**:

Gewers FL, Ferreira GR, de Arruda HF, Silva FN, Comin CH, Amancio DR, Costa LD. Principal component analysis: A natural approach to data exploration. ACM Computing Surveys, Volume 54, Issue 4, Article No.: 70, pp 1–34, https://doi.org/10.1145/3447755

Isokääntä, S., Kari, E., Buchholz, A., Hao, L., Schobesberger, S., Virtanen, A., and Mikkonen, S. (2020)
Comparison of dimension reduction techniques in the analysis of mass spectrometry data
Atmos. Meas. Tech., 13, 2995–3022, doi:10.5194/amt-13-2995-2020.

---

## Referee Comment (RC2)

Review of Stamatis et al.: "Development and Application of a Supervised Pattern Recognition Algorithm for Identification of Fuel-Specific Emissions Profiles"

**General Comment:**

The researchers trained an unsupervised machine learning model utilizing PCA, k-means clustering, and LDA to identify fuel types by their monoterpenoid combustion emission profiles. They then apply this classification model to synthetic mixtures and to real field data. They find performance of the PR scheme declines as synthetic mixtures come closer in composition, but that in the case of field data, performance was good despite the presence of unknown fuel types.

Basically, there just needs to be a bit more separation and depth into the methodology given the complexity of it. All methodology should be explained up front and then results considered. More depth on methods would mean things like why PCA was the decided dimension reduction technique, and why use only 2 PCs as input to your cluster analysis (would this have led to overfitting of the data?).

Nonetheless, using machine learning to build intricate fuel type fingerprints is very interesting and I can recommend this work for publication following these minor revisions.

**Specific Comments:**

Line 151: Is the LOD mentioned here the same as the "detection limit" mentioned in line 115? What is this detection limit specifically? Is this in reference to the instrumentation LOD used across these different campaigns?

Line 161: What's the rationale behind performing the manual selection at all? Is the idea that this demonstrates that your feature selection is more sophisticated than effectively just guessing?

Section 3.2.1, Line 177: Fig. 2 seems unclear to me in what it's describing. My understanding is having fewer PCs describe more variance indicates better separation, but I think that could be made clearer (or perhaps I am altogether wrong). For instance, why mention only PCs 1-4 in line 178, why is PC5 not considered? Why are PCs 3-4 mentioned when only PCs 1-2 are used in the PR model?

Line 181: The automated criteria result in "more distinct and more consistent fuel profiles for each family," but I don't quite see how that's exemplified in Figures 3 and 4.

Line 250: What is $C_{st}$ in eqn. 4?

**Technical Corrections:**

Line 56: "…including to identify" should be "…including the ability to identify"

Line 92-93: Add comma, "…FIREX FL16, a broad…"

Line 119: Add comma, "In this work, an analysis of variance…"

Line 155: Add comma, "During the FIREX-16 FL16 study, …"

Line 170: Add comma, "In this application, …"

Line 192: Change "from the retained PCs (PC1 and PC2)" to "from these retained PCs"

Line 195: Add comma, "In this study, …"

Line 317: Remove comma after "algorithm"

Line 336: Add "of" to "27 out 28"

Figures 14-16 would be better conveyed were they vertically stacked.

---

## Author Comment (AC1)

Here we have prepared responses to the three reviews of our submitted manuscript. Two of the reviews were focused largely on the details of the statistical approaches, and the other on the broader implications of these tools for biomass burning emissions and modeling. All three of the reviewers suggested structural changes to improve clarity. We thank each of the reviewers for their comments, questions, and suggestions which have served to improve the manuscript. Our responses are in blue, and modifications to the text are in red. Line numbers refer to the version of the manuscript showing tracked changes.

**Reviewer # 1**

**General Comments:**

The manuscript introduces interesting algorithm for identification of different fuel types from emission profiles. The algorithm is a combination of widely applied statistical methods which together make a good tool for emission classification. However, the methods need to be described with more detail and the manuscript needs to be structured better. The methods should be separated from the results, now the text is difficult to read as new methods pop out from nowhere in the middle of results. With these and the detailed comments below properly addressed I could recommend the manuscript for publication.

In response to the general comments provided by Reviewer #1, we have expanded the methods description and better separated the methods from the results and discussion. The manuscript is now structured as follows: Section 1 (Introduction), Section 2 (Data and Methods), Section 3 (Results and Discussion), and Section 4 (Conclusions). All the methods are now described in Sections 2.2 - 2.4; additional detail has been added in these sections and in the SI, Sections 2-4.

**1. Page 4 lines 115-118**: Is the reason for missingness always the compounds being below the detection limit? If there are missingness due to e.g. instrument malfunction, filling in the zeros might bias the further analysis.

In this study there were two reasons why values were missing: 1) compound levels were below the detection limit (0.1 ng); or 2) compound levels were negative after background subtraction. In response to this comment, we have modified the manuscript (lines 114-121) to acknowledge that replacing missing values should reflect the type(s) of data and the reason(s) for missing values, and that the approach applied here is specific to the data presented. The text now reads:

"Data preprocessing (step 1) is performed to handle any missing values in the samples. Approaches for handling missing values are specific to the type(s) of data and the reason(s) for missing values (Dong and Peng (2013); McNeish (2017)). In this data set, missing values largely are a result of compounds being below the detection limit or having negative values after background correction. During preprocessing, for every feature (i.e., compound) the percentage by number of missing values across all samples was calculated. For any given compound, if the percentage of missing values was less than 30% then the missing values were replaced with zeros. If the percentage was more than 30% then the compound was removed from the data set. The 30% threshold is supported by published statistical methods guidance studies including Dong and Peng (2013) and Jakobsen et al. (2017), that suggested a threshold range of 10% - 40% prior to replacement. " **2. Page 4, lines 119-122**: Feature selection needs a bit more clarification. ANOVA is sensitive to nonnormality and heteroscedasticity of the data and if this is not taken account the feature selection may be biased. Whereas for PCA, standardization should not be standard procedure (Gewers et al. 2018), especially if the components are calculated with singular-value decomposition (SVD, Isokääntä et al. 2020) as I assume was applied here since the authors refer to Abdi & Williams 2010 paper. In addition, it should be stated in the methods section if SVD or eigenvalue decomposition was used to find the principal components.

We acknowledge the reviewer's comment that ANOVA can be sensitive to nonnormality and heteroscedasticity. The EF data used in this study did not have a normal distribution. Regarding heteroscedasticity, after preprocessing, some of the remaining compounds had unequal variances while others did not. Nonnormality is particularly a problem when the sample size is small (below 15-20 observations depending on the study); this is a concern for our data and needs to be explored in future work. We have added a sentence in the conclusions (lines 439-442) that acknowledges this point. The text now reads in lines 439-442:

**"Future work should include exploring how normality and heteroskedasticity, which are underlying assumptions for ANOVA, may be affecting separation. This can be achieved by using non-parametric tests that do not make assumptions about the underlying data distribution and are more robust than ANOVA in the presence of heteroskedasticity."**

These deviations likely would result in fewer compounds being selected; i.e., some compounds that may be statistically different between fuels are not identified using ANOVA. We explored this possibility by increasing the number of selected compounds and evaluating the separation of fuels for five compounds (Fig. 6) and for nine compounds (Fig. S6). Overall, more compounds did not provide better separation. All pine samples clustered on their own with no overlap, spruce had only one overlapping sample with firs, and firs had one overlapping sample with spruce.

The reviewer is correct that SVD was used to find the principal components. We have added this detail to the manuscript (lines 152-154):

**"The PCs can be calculated either by eigenvalue decomposition (EVD) on the covariance matrix of the original data or singular value decomposition (SVD) on the data matrix. SVD was used in this study..."**

We thank the reviewer for the Gewers et al. (2018) reference. We acknowledge that standardisation prior to PCA needs careful consideration and should not be a default procedure. In this application of the algorithm, the variance in the emissions was attributed to differences within and across fuel types and not to instrumental artifact or other intrinsic variability associated with the analytical method. However, because it is our intention to make the algorithm generally applicable, we have modified the manuscript (lines 158-162) to reflect these points. The text now reads:

"While standardization can help alleviate the scaling problem it should not be a default practice as it can magnify the effect of outliers in the data (Gewers et al. (2021)). If the variability of a feature is a consequence of intrinsic variability in the analytical method (e.g., experimental error or noise in the data), then standardization may erroneously emphasize

**that in the PCA results. In such cases, either the noise should be reduced by some means or standardization should be avoided."**

**3**. The whole section introducing the PR algorithm should be extended to give more details on the methods used. It should be stated why PCA was chosen over other dimension reduction methods, like exploratory factor analysis, and similarly for LDA. PCA execution should be described with more details. In addition to the decomposition method, it should be stated if some rotation method was used and which type of rotation: orthogonal or oblique and which version of them. Rotation methods are discussed in Abdi & Williams and in Isokääntä et al. (2020).

Regarding the selection of PCA, during our literature review for pattern recognition analysis we identified several studies that met similar research objectives using PCA. Two particular studies influenced this project, Welke et al. (2013) and Ziolkowska et al. (2016), because these studies focused on separation of samples using one- and two-dimensional gas chromatography data. In response to this comment, the manuscript has been modified (lines 162-164) and now reads:

**"In this study, the selection of PCA for dimensionality reduction was based on three previous studies that included PCA in pattern recognition analysis of chromatographic data (Welke et al. (2013), Johnson and Synovec (2002), Ziółkowska et al. (2016))"**

Regarding classification, LDA was chosen because of its closed-form solution. While other classification methods could have been used, such as k-Nearest Neighbours (kNN) or Support Vector Machines (SVMs), they require tuning specific hyperparameters. In this study one of the main objectives was to train an algorithm on laboratory data and test how it performs on field data. While kNN or SVMs could have been trained on laboratory data using *k*-fold cross validation: 1) the hyperparameters would have been optimised for laboratory data only and 2) we did not have labels for the field samples like we did for the laboratory samples. Thus, methods that require efficient hyperparameter tuning were deemed not optimal for this case study. In response to this comment, the manuscript has been modified, including a statement on rotation (lines 216-219), to read:

**"LDA was chosen as the classification method in this study because of its closed-form solution that does not require any hyperparameter tuning. Methods that require hyperparameter tuning (e.g., *k*-Nearest Neighbours) are not appropriate for this application because: 1) the training data set included laboratory data only while the test set included field data, and 2) the field data were not labeled."**

**4. Section 3.2.1.** The feature selection needs more justification. As PCA is, by definition, dimension reduction method, why the dimensions need to be reduced before running PCA for PR? Was the explained variance used as selection criteria for lower number of compounds in the analysis? It should not be as higher number of variables means higher total variance, which in turn leads to lower explanation rate with same number of components. Is the result in Fig 2. for the selected 5 (or 10) variables? I yes, then it should be noted that with five components you will explain 100% of the variation of five variables (but not for 10 variables). Since PCA tries to explain most of the variance with the first component, it is expected that the explanation rate is high. Naturally, with statistical models the aim often is to get as simple model as possible but the selection criteria need to be well reasoned.

While we acknowledge that PCA reduces the dimensionality of the initial data, it does so in an unsupervised way without taking into consideration which features would be more appropriate for class separation. We know that in the full data set we have include variables that do not provide information for class separation. By introducing the ANOVA selection before PCA, the potentially informative features for class separation (in the PCA space) can be detected much faster.

We thank the reviewer for this point regarding the number of components, number of variables, and explained variance. The explained variance was used as an indication of effective dimensionality reduction using only principal components based on the selected variables (Fig. 2). In response to this comment, we have revised Fig. 2 to show cumulative explained variance for three test cases: manual selection (9 compounds), automated selection (5 compounds), and automated selection (9 compounds).

**5. Line 189. Fig 2 does not show normalised eigenvalues. Add the scree plot**

Thank you for pointing this out. We have added the scree plot for both the automated and the manual selection in the SI, Figs. S2-S4.

**6. Line 195:** Why Euclidean distance? I am not questioning the choice but wish to see the reasoning.**

The reported use of the Euclidean distance in the manuscript was an error. Instead, the total within sum of squares (TWSS) was used as a distance metric. TWSS was chosen because the *k*-means algorithm tries to minimize the same metric for each cluster. Therefore using the same metric to evaluate the overall error was deemed appropriate. The manuscript has been updated (lines 185-187) in response to this comment and now reads:

**"The elbow plot method requires running the k-means multiple times using a different number of clusters each time. For each run the total within-sum of squares (TWSS) was calculated according to Eq. 4:"**

**All other references to Euclidean distance were replaced with TWSS.**

**7.** Section 3.2.3: Did I understand correctly that you were only using one pair of components at the time in cluster analysis? If yes, I would ask why? Cluster analysis is a multivariable method, and thus can be applied even for all components at the same time.

We acknowledge the statement by the reviewer that *k*-means is a multivariable method and can accommodate all components at the same time. One of the goals of this application was to achieve separation of the fuel types with as few compounds and components as possible, while maintaining more than 80% explained variance after PCA and minimum overlap between fuel types. There are experimental design advantages to being able to identify fuels using a highly reduced set of measured compounds (e.g., see also responses to Reviewer #3). With the selected compounds (feature selection approach) every combination of PC1 with any other component yielded more than 80% explained variance. Therefore running k-means with more than two components would be redundant, provided that we achieved the fuel separation that we did with the combination of (PC1, PC2) and (PC1, PC4). In the updated manuscript we have added a section (see Section 2.3) where we explain in more detail how the principal components are used in conjunction with the k-means algorithm. The

text now reads:

**"2.3 Running the pattern recognition algorithm**

Implementation of the PR algorithm proceeds through five steps. In step one, the compounds in the data set are preprocessed to replace missing values and discard samples that might be problematic (section 2.2.1)..."

**8.** Figure 8 shows that PC4 does not really affect the clustering, but it is defined by PC1. Thus, this paired comparison seems redundant.

We agree with the reviewer that most of the separation takes place along PC 1. Though by comparing Figs. 6 and 8 it can be seen that while PC1 accounts for most of the separation, combined with PC 2 (Fig. 6) it does not result in complete separation of the firs from spruce (e.g., four spruce samples overlap with firs Fig. 6). Using PC1 and PC4 (Fig. 8) we increased the separation between firs and spruce with only one spruce sample overlapping with firs.

**9. Page 13, line 243-244**: Separation would be improved also by increasing the number of compounds in the analysis or using different rotation in PCA.**

For this application, our analysis suggests that increasing the number of compounds would not necessarily improve separation. In response to this comment, we provide one example in which we compared the separation achieved using nine selected compounds (PCA plot, Fig. S6) with that achieved using the five selected compounds. The extra four compounds provided similar results with five compounds (PC1 and PC4) but only with PC1 and PC2. Specifically the clustering results in 16/16 pines clustering in their own group (SI S6 cluster 1), two spruce clusters overlapping with two fir samples (S6 clusters 3 & 4), and firs overlapping with one spruce sample (S6 cluster 2). Furthermore the total explained variance (by PC1 and PC2) was about 70% which was 10% less than the 80% set threshold and about 12 % less that the explained variance by PC1 and PC4 with five compounds. At the same time we wanted to achieve the separation with the simplest emission profile as possible (i.e., as few compounds as possible). Furthermore we wanted to avoid potentially overfitting the model by adding too many compounds. Overall the fuel separation with the five compounds and no rotation was deemed adequate for this study. We acknowledge that future applications may have better results using more compounds/features.

**10.** Results and Conclusions: Too strict limitations in feature selection also reflect results and conclusions. Using more features in the model would probably lead to better separation of sources in mixed samples but increasing the number too much would decrease generalizability of the model or even lead to overfitting. Thus, it is important to find the balance.

We agree. With repeated application, and expansion to different data sets, we will be able to provide further recommendations regarding feature selection as a function of the size of the data set and goals of the application.

**Reviewer # 2**

**General comments:**

The researchers trained an unsupervised machine learning model utilising PCA, k-means clustering, and LDA to identify fuel types by their monoterpenoid combustion emission profiles. They then apply this classification model to synthetic mixtures and to real field data. They find performance of the PR scheme declines as synthetic mixtures come closer in composition, but that in the case of field data, performance was good despite the presence of unknown fuel types. Basically, there just needs to be a bit more separation and depth into the methodology given the complexity of it. All methodology should be explained up front and then results considered. More depth on methods would mean things like why PCA was the decided dimension reduction technique, and why use only 2 PCs as input to your cluster analysis (would this have led to overfitting of the data?). Nonetheless, using machine learning to build intricate fuel type fingerprints is very interesting and I can recommend this work for publication following these minor revisions.

These general comments from Reviewer #2 are broadly consistent with those from Reviewer #1. We have restructured the manuscript such that the methods are better separated from the results and discussion, and have provided additional details on the methods themselves. Please see responses to comments to Reviewer #1 for additional details.

**1. Line 151**: Is the LOD mentioned here the same as the "detection limit" mentioned in line 115? What is this detection limit specifically? Is this in reference to the instrumentation LOD used across these different campaigns?

Yes. We thank the reviewer for catching this inconsistent use of terminology. The detection limit is in reference to the instrumentation used for the field and laboratory measurements, specifically two-dimensional gas chromatography with time-of-flight mass spectrometry. We have modified the manuscript so that only "detection limit" is used and in the first appearance the mass threshold (0.1 ng) is reported.

**2. Line 161**: What's the rationale behind performing the manual selection at all? Is the idea that this demonstrates that your feature selection is more sophisticated than effectively just guessing?**

In our previous efforts, differences in emissions between fuel types were visualized by performing standard data analysis procedures (e.g., checking NIST library matches peak by peak; assigning each peak a name (if possible), structure (if possible), formula, and compound class) and plotting emission factors as a function of carbon number and functionality (Hatch et al. (2015), (2017)), plotting peak abundance correlations (Hatch et al. (2018)), and/or plotting fractional composition (Hatch et al. (2019)). It was clear from these analyses that fuel type, and not just combustion efficiency, was resulting in non-negligible differences in emissions. A logical extension of that work was to select compounds that may be useful in differentiating the fuel(s) that contribute to any given smoke sample (i.e., "manual selection"). However, the large number of compounds (hundreds) suggested that manual selection may not be as effective as a machine learning approach. Further, manual selection is extremely time consuming. These observations led to the development and testing of the algorithm presented in this work. Thus the reasoning for presenting a comparison between the manual and automated feature selection was two-fold: 1) to determine whether the automated feature selection produced results that were consistent with Hatch et. al (2019); and 2) to evaluate whether the algorithm/automated feature

selection was in fact more effective than manual selection and added information not previously obtained through more traditional data analysis

**3. Section 3.2.1, Line 177**: Fig. 2 seems unclear to me in what it's describing. My understanding is having fewer PCs describe more variance indicates better separation, but I think that could be made clearer (or perhaps I am altogether wrong). For instance, why mention only PCs 1-4 in line 178, why is PC5 not considered? Why are PCs 3-4 mentioned when only PCs 1-2 are used in the PR model?

The reviewer is correct that having fewer PCs describing a higher % variance indicates better separation. In response to this comment, as well as comments from Reviewer #1, Fig. 2 has been revised and Section 2.3 has been expanded to better describe the observed relationships between feature selection, number of PCs, and explained variance. Please also see the responses to comment 4 by Reviewer #1. Regarding PCs 1-4, this was an error and should have been PCs 1-5. We thank the reviewer for catching this. Furthermore PC1 and PC2 are not the only ones used. We evaluated all combinations of PC1 with the rest of the components. The evaluation showed that the combinations of (PC1, PC2) and (PC1, PC4) were important to separate the three fuel types and that (PC1, PC4) was important to further separate spruce (see response to comment 7 from Reviewer #1).

**4.** Line 181: The automated criteria result in "more distinct and more consistent fuel profiles for each family," but I don't quite see how that's exemplified in Figs. 3 and 4.**

In Figs. 3 and 4, consistent fuel profiles are demonstrated by the similarity in the normalized emission ratios among the fuel types: firs, pines, or spruce. For example, based on the manual selection criteria, the selected compounds a-pinene, d-limonene, and b-phellandrene have very different relative abundances between the lodgepole and ponderosa pines (Fig. 3). Because these same compounds were not selected based on the automated selection criteria, these differences are not apparent in Fig. 4, and the relative abundance profiles are much more similar for the pines in Fig. 4 than in Fig. 3. Further in addition to the emission ratio profiles being more similar among the types in Fig. 4, the emission ratio profiles are also more different between the fuel types, demonstrating the distinction realized by the automated selection criteria. For example, the emission ratio profiles for lodgepole pine and Engelmann spruce are arguably more similar in Fig. 3 than the profiles for lodgepole and ponderosa pine. However, in Fig. 4, the profiles for lodgepole and ponderosa pine are more similar and the pines are distinct from the spruce. We have revised the colour scheme in the figures, including Figs. 3 and 4, which may help improve clarity. In addition we have revised the text in Section 3.2.1 (lines 272 -274) to read as follows:

"Emerging patterns can be seen in the resulting profiles between and within the fuel types. The emission profiles from the automated selection, for both five and nine compounds, provide more distinct profiles between fuel types, and more consistent profiles within types, than the profiles from manual selection..."

**5. Line 250: What is Cst in eqn. 4?**

Cst includes constants from the multivariate Gaussian distribution which is assumed during the derivation of Eq. 5. We updated the manuscript to reflect that in line 215:

"Cst is a term that contains constants from the multivariate Gaussian distribution"

In addition, we addressed the derivation of Eq. 5 in more detail in the updated SI (Section 5).

**Technical Corrections**

**Line 56**: "...including to identify" should be "...including the ability to identify" Done. Thank you.

Line 92-93: Add comma, "...FIREX FL16, a broad..." Done. Thank you.

Line 119: Add comma, "In this work, an analysis of variance..." Done. Thank you.

Line 155: Add comma, "During the FIREX-16 FL16 study, ..." Done. Thank you.

Line 170: Add comma, "In this application, ..." Done. Thank you.

**Line 192**: Change "from the retained PCs (PC1 and PC2)" to "from these retained PCs" Done. Thank you.

Line 195: Add comma, "In this study, …" Done. Thank you.

Line 317: Remove comma after "algorithm" Done. Thank you.

Line 336: Add "of" to "27 out 28" Done. Thank you.

Figures 14-16 would be better conveyed were they vertically stacked Done. Thank you.

**Reviewer # 3**

**1.** The scientific premise of the paper - to use measured NMOC profiles to predict the underlying burned fuel - is interesting and worth publishing (although from a modeling perspective, perhaps the inverse would be more useful). However, the paper is so poorly written and organized it is difficult to discern what algorithm the authors have developed, and it would surely not be reproducable by a reader. I recommend to reconsider this paper after major revisions.

Regarding the use of NMOC profiles to predict underlying burned fuel, and the utility of the inverse, it has been our experience that accurately identifying burned fuels can be very challenging in the absence of detailed pre- and post-burn surveys. This is due to a number of factors, including the difficulty of maintaining up-to-date fuel loading maps. Thus part of the motivation for this work was the potential for identifying burned fuels using a reduced set of NMOCs that were likely to be routinely measured in the field, which then could be linked back to more detailed speciation profiles for modeling.

Regarding the organization of the paper, Reviewers #1 and #2 expressed similar concerns; the paper has been reorganized to more clearly separate the methods from the results and discussion, and additional details on the methods have been added. Please see responses to comments to Reviewer #1 for additional details. Regarding reproducibility, we also note the availability of scripts and examples in the referenced GitHub library.

**2.** The authors do not report what NMOC compounds make up their training and testing data sets. At some point, they do mention that there are 93 to select from. They should be listed and described. The authors write about generating a synthetic data set, but no information was given on how this data was generated. The laboratory burn data set used included a broad varity of biomass fuels, but in the end only a hanful of fuels were considered, and no explanation was given, which calls into question the extent of the applicability of the algorithm.

In response to this comment, we have added a new section to the SI (Section 1) in which we report on a similarity analysis using the 458 detected NMOCs from the FLAME-4 FL12 data set, which highlights the importance of monoterpenes in separating coniferous fuels. We focus on coniferous fuels due their importance in the Western US (a point of focus for the FIREX-AQ campaign) and the availability of field data from consumption of these fuels. Fuel types that had more than 30% by number of the 93 reported monoterpenes were retained (see line 118). The 93 monoterpenes are reported in Hatch et al. (2015), as referenced in lines 103 - 105 of the manuscript. After the data processing steps, 12 monoterpenes in 39 samples remained. The selection of these monoterpenes and these samples ensures that the necessary statistical properties can be calculated and that a representative number of samples are available in each fuel type (described in section 2). The retained monoterpenes are now listed in Table 2 in the SI.

Regarding the synthetic data set, we generally wanted to test the algorithm on mixed fuel samples, as is expected in the field. Therefore, the synthetic data set was comprised of simplified mixtures of two fuels assuming linear combinations at fixed percentages of each fuel. We have updated the manuscript in lines (228-229) to reflect that and not it reads as:

"The synthetic samples were created by taking the average value of each selected NMOC from each fuel type and linearly combining it using specific percentages for each fuel (eg. 60/40, and 90/10)"